# The Influence of Healthy Lifestyle on Willingness to Consume Healthy Food Brands: A Perceived Value Perspective

**DOI:** 10.3390/foods14020213

**Published:** 2025-01-12

**Authors:** Elizabeth Emperatriz García-Salirrosas, Jorge Alberto Esponda-Perez, Dany Yudet Millones-Liza, Karla Liliana Haro-Zea, Luiggi Agustin Moreno-Barrera, Ghenkis Amilcar Ezcurra-Zavaleta, Luis Alberto Rivera-Echegaray, Manuel Escobar-Farfan

**Affiliations:** 1Research and Innovation Group for Entrepreneurship and Sustainability, Universidad Nacional Tecnológica de Lima Sur, Lima 15816, Peru; 2Faculty of Nutrition and Food Sciences, Universidad de Ciencias y Artes de Chiapas, Tuxtla Gutiérrez 29000, Mexico; jorge.esponda@unicach.mx; 3Unidad de Ciencias Empresariales, Escuela de Posgrado, Universidad Peruana Unión, Lima 15102, Peru; dannie@upeu.edu.pe; 4Facultad de Ingeniería, Ciencias Administrativas y Sociales, Universidad Autónoma de Baja California, Sede Tecate, Mexicali 21100, Mexico; kharozea@gmail.com; 5Faculty of Management Sciences and Communications, Universidad Autónoma del Peru, Lima 15842, Peru; lmoreno9@autonoma.edu.pe; 6Facultad de Ciencias Económicas, Escuela de Administración, Universidad Nacional de Tumbes, Tumbes 24001, Peru; gezcurraz@untumbes.edu.pe; 7Universidad Tecnológica Peru, Lima 15046, Peru; lriverae@utp.edu.pe; 8Department of Administration, Faculty of Administration and Economics, Universidad de Santiago de Chile, Santiago 9170020, Chile; manuel.escobar@usach.cl

**Keywords:** healthy lifestyle, healthy foods, perceived value, purchase intention, healthy brand marketing, Peruvian consumers

## Abstract

This paper aims to build a predictive model that assesses how a healthy lifestyle affects different dimensions of perceived value (quality, social value, emotional value, and economic value) and how these dimensions impact the willingness to purchase healthy brands. A quantitative, non-experimental, and cross-sectional study was conducted with a sample of 515 participants. A self-administered questionnaire was used, and the data were analyzed using the PLS-SEM method. The findings indicate that a healthy lifestyle positively influences the perception of quality and the social, emotional, and economic value of healthy brands. However, only perceived quality, social value, and emotional value significantly impacted the willingness to purchase, while economic value did not show a relevant effect. It is concluded that a healthy lifestyle is a determining factor in the perception of the value of healthy foods, which reinforces the purchase intention, except for economic value. These findings suggest that companies should prioritize strategies that reinforce the quality and emotional and social connections of their products to encourage the consumption of healthy brands. This study contributes theoretically to the perceived value model in emerging markets and provides practical implications for designing more effective marketing strategies in the Peruvian context.

## 1. Introduction

A healthy lifestyle has become a trend of global importance [1,2]. Issues such as preventing excess weight, obesity, and related diseases promote the consumption of healthy products that promote this way of life [3]. In that sense, a lifestyle is people’s general way of living, which arises from the interaction between living conditions in a broad sense and individual behavior [4]. Now, when it comes to defining a healthy lifestyle, it entails having good continuous health, which implies eating well, sleeping well, exercising, managing stress, abstaining from drug use, and relying on social support [5,6]. This can individually improve people’s longevity [7]. This reformulation of a healthy approach to food is a potential way to improve the quality of diet in homes [8,9]. Therefore, the communication companies make towards consumers with a healthy lifestyle, such as emotional marketing, positively impacts the purchase intention of healthy food products [10,11]. For example, when it came to communicating changes in the content of healthy food products, the use of eco-friendly packaging, nutritional claims relating to low-calorie intake, low salt, and high macronutrient products, and the use of opinion leaders with whom consumers can identify, all took on importance in the purchasing decisions of people seeking to achieve a healthy lifestyle [12]. In a study carried out by Diabetes UK on 2121 adults, it was found that nutritional traffic light labeling increased the perception and intention to purchase healthy products by more than 29% compared to a zero healthy labeling system [13].

On the other hand, in a study carried out in the retail sector, it was found that a healthier distribution in supermarkets can improve the purchases of people seeking to obtain and maintain a healthy lifestyle, and it also suggested considering placing fruits and vegetables near the entrances [14]. The same thing happened with the virtual customer, who, when scanning the products on his or her cell phone when preparing to purchase any product, received digital messages called visibility pushes, alerts of healthy product qualities, which improved the perception regarding their value and consequently influenced his or her purchase [15]. That is, personal beliefs and social norms are elements that influence consumers’ attitudes and perceptions [16,17]. People with a positive perception of knowledge of healthy nutrition achieve a correct acquisition of nutritious food products [18]. In this sense, the present study raises the following research question: What is the influence of a healthy lifestyle (HLB) on the perceived value of quality (PQV), social value (PSV), emotional value (PEV), and economic value (PFV) in the willingness to consume healthy food brands (WCFHB) in the Peruvian market?

In this way, the main objective of this study is to explain two things: first, to propose a theoretical model where the healthy lifestyle (HLB) influences the dimensions of perceived value, such as quality perception (PQV), social value (PSV), emotional value (PEV), and economic value (PFV). That is, to prove that having a healthy lifestyle can contribute to the key factors of perceived value, such as perceived quality, social value, emotional value, and economic value. Secondly, to determine the influence of these four variables on the willingness to consume foods from healthy brands (WCFHB) in the Peruvian market, considering the implications they may generate in the economic, social, and business spheres. Research on the analysis based on the theory of perceived value of healthy food products has significant relevance in people’s willingness to purchase a healthy lifestyle. The results of this research can provide valuable information about the desire to purchase healthy products based on perceived value, generating a positive impact on the marketing of healthy products and contributing to the progress of a society with a healthy lifestyle.

The structure of this paper is as follows: Section 2 presents the literature review and the theoretical model, Section 3 presents the materials and methods, Section 4 presents the results, and Section 5 presents the discussion, implications, future research, and, finally, the conclusions.

## 2. Literature Review

Perceived value refers to the evaluative judgment of an individual at the time of acquiring a product or service; this is carried out according to the perception regarding the result obtained [19] and the evaluation of its usefulness based on the benefit obtained versus the price paid for the product or service [20]. For a better understanding, various studies explain the categories that are part of the perceived value, among them social value, which refers to the symbolic acts of a brand whose fundamental basis is improving the social and environmental conditions of the environment. This action allows the client to exhibit positive behavior, increasing the purchase frequency of brand products [21,22]. Another category is emotional value, which, according to previous studies, is the sentimental bond between the brand and the consumer. It is clear that when it comes to products that contribute to a better lifestyle, emotional value generates security and relevance, becoming an added value for the brand because it can turn its consumers into its defenders, thus promoting retention and recommendation [21,23].

Another relevant category is financial value, which represents a tangible aspect. Consumers demonstrate their willingness to pay based on their perception of the product and evaluate their payment based on the benefits they can obtain [24,25]. Research has shown that this factor is one of the key determinants of food purchase intention [26]. Similarly, perceived quality is an essential component that examines customers’ subjective perception of a product’s excellence or superiority compared to another. This aspect can be measured through individual consumer evaluations [27,28].

A healthy lifestyle is the regulation of the behavior that an individual acquires regarding the contribution to his or her physical, mental, and social well-being [29,30] and the decision to adopt habits and structure daily activities so that these actions help prevent diseases, increase longevity, improve quality of life, and increase life expectancy [31,32]. Regarding the willingness to consume healthy foods, it is probable that a consumer will adopt a behavior based on their motivations, beliefs, and food perspectives towards a particular product [33]. This behavior depends on the confidence acquired by the product in the market since, in modern times, there is a favorable attitude towards healthy foods [34,35].

### Development Hypothesis

Every action associated with the daily behavior choice of an individual has a significant impact on their physical, mental, and/or social health. These actions are called lifestyles, as they are part of a pattern that influences the perceived quality of life. How individuals value their well-being influences their expectations, standards, and concerns [36]. In addition, it is known that perceived quality can be evaluated according to the conduct of an individual’s lifestyle, and according to history, an unhealthy lifestyle minimizes the value of the quality of a particular product. This is due to the behaviors and habits that affect their perception regarding the attributes that a product could offer [37]. Even a young person with a sedentary life could care little about the nutritional quality of a product because they do not feel an immediate need to give importance to a healthy product that improves their health condition [38,39]. This means that when people adopt a healthy lifestyle, they maintain high expectations when choosing a product and mostly resort to choosing healthy brands as a food safety measure [40,41]. In light of the above, the following hypothesis is established: 

**H1.** 
*Lifestyle healthy behavior significantly influences the perceived quality value of healthy food brands.*


This considers that lifestyle involves daily intake and food selection, and also affects the assessment of a product based on its environmental values and the acquisition of cruelty-free options. For populations that maintain a healthy lifestyle, food decisions are based on evaluating their contribution to health and includes the commitment to ethical and sustainable purchases [42,43]. Likewise, the background refers to a group of people with a healthy and sustainable lifestyle personality (LOHAS) who, by their perception, tend to prefer food recognized for respecting the environment or those called ecological [44,45]. In general terms, consumers whose lifestyles are healthy and are recognized as health promoters maintain a positive behavior towards caring for the environment, support a sustainable and ethical vision in their consumption choices, and seek to acquire products that minimize the use of resources [46]. This type of population presents a value orientation that contributes to everything oriented towards society’s progressive development [47]. Under the above, the following hypothesis is established:

**H2.** 
*Healthy lifestyle behavior significantly influences the perceived social value of healthy food brands.*


Previous studies have established that people’s lifestyles are related to a specific pattern and a concern for their health [7,48]. Consequently, when they identify a product, which from their perspective, contributes to the stability of their health, it generates a satisfactory experience that opens the door to a feeling of connection, which can increase until this generates a positive cycle of consumption towards a particular product; in this way, the literature supports the fact that, for the population that maintains a good lifestyle, the emotional value becomes one of the most relevant factors, this is because together with the nutritional criterion, people add the experience of pleasure and connection with a particular product or brand [44]. This means that practical values are highly valued by people who study healthy living since they appreciate things that positively affect economic growth and the conservation of natural resources. Beyond consuming healthy products, the population with a healthy lifestyle creates a unique connection of identity and relevance that goes beyond a joint purchase [23]. Considering the above, the following study hypothesis is proposed:

**H3.** 
*Lifestyle healthy behavior significantly influences the perceived emotional value of healthy food brands.*


Within the area of public health, various threats have been identified that put people’s health at risk; in the face of this, the literature establishes that adopting a healthy lifestyle is a significant alternative to increase life expectancy and improve the quality of life of a person. This is apart from the associated costs in that, when it comes to a person’s health, financial resources lose priority, driving the intention to invest more in exchange for having a better health condition [31]. Studies support the idea that economic value is better valued by those who maintain an adequate lifestyle: they are more willing to pay because they know and value the benefits of their food choice [49]. In addition, this population often allocates a defined budget to acquire healthy products, so money does not represent a barrier when buying a product from a healthy brand [50], since the priority of those who maintain an adequate lifestyle is their health, even when this means paying a high price [33,51]. Based on the findings, the following hypothesis is proposed:

**H4.** 
*Lifestyle healthy behavior significantly influences the perceived financial value of healthy food brands.*


Research shows that quality perception influences purchase intention and is linked to brand equity [52]. Their research indicates that better quality perception can increase the desire to purchase products, which is essential in healthy food, where consumers prioritize alternatives that promote their well-being. Similarly, other previous research mentions that labels that highlight healthy attributes (such as “low fat” or “gluten-free”) impact quality perception [53]. In this context, clear and accurate information can influence consumer behavior and is relevant to the food industry in a broad sense. Labeling that correctly communicates health benefits can improve the perception of quality and increase purchase intention [54]. Similarly, researchers propose that the perception of a product’s effectiveness can affect the intention to make responsible consumption choices [55]. This suggests that consumers who perceive high-quality, healthy products are more inclined to choose them instead of less healthy alternatives. This approach coincides with the idea that perceived quality can indicate consumption intention, particularly in products that promote a healthy lifestyle. Also, evidence supports the fact that the perception of the intrinsic quality of certain products, such as fresh cherries, positively affects consumer satisfaction and, consequently, their purchase intention [56]. In light of the above, the following hypothesis is established: 

**H5.** 
*Perceived quality value significantly influences the willingness to consume healthy food brands.*


Human values, such as universalism and benevolence, affect the consumption of food products. These values are related to the intention to purchase foods that meet personal needs and promote social and environmental well-being [57]. Meanwhile, researchers have examined how social groups influence the food decisions of college students, showing that friendships and social circles can exert considerable pressure on their consumption choices [58]. Similarly, a study on brand value and its link to purchase intention highlights those social factors, such as word-of-mouth recommendations, in determining buying decisions [59]. This indicates that consumers who perceive a high social value in healthy brands, either through brand communication or the influence of their peers, will be more likely to opt for these products. Additionally, the study supports the notion that perceived social value and other emotional and economic values influence purchase intention. Thus, those consumers who believe their food choices benefit society tend to prefer healthy brands [60]. Based on the above, the following hypothesis is established:

**H6.** 
*Perceived social value significantly influences the willingness to consume healthy food brands.*


Taste perception and the generation of emotions influence food selection. It has been proven that olfactory perception, capable of awakening memories and emotions, affects the decision to consume food, even unconsciously [61]. A recent study indicates that attitudes towards a healthy brand impact consumers’ purchase intention, suggesting that a favorable emotional connection can facilitate the selection of healthy products [62]. This indicates that emotions linked to certain foods can be crucial in consumption intention, particularly in healthy foods. Perceived emotional value has been identified as a critical factor in predicting purchase intention in various contexts. For example, among Indian consumers, emotional value influenced purchase intention for local brands significantly, indicating that consumers prioritize emotional connections over functional attributes when making decisions [63]. Similarly, a study on Mexican students revealed that emotional value is a significant predictor of purchase intention for apparel brands, reinforcing the importance of emotions in forming purchase intentions [64]. In Brazil, they identified that perceived emotional value positively influences trust and purchase intention, underlining the relevance of subjective factors, such as pleasure and well-being, in decision-making related to food purchases [65]. In light of the above, the following study hypothesis is proposed:

**H7.** 
*Perceived emotional value significantly influences the willingness to consume healthy food brands.*


On the other hand, it has been identified that the perception of economic value directly impacts the intention to purchase, suggesting that consumers consider the cost of the perceived benefits of healthy products when making their purchasing decisions [65]. Specifically, in Ecuador, consumers express interest in the quality of organic products. However, they are reluctant to pay higher prices, which reveals a discrepancy between perceived economic value and purchase intention [66]. Evidence has also been found to support the influence of social groups on the diet of university students in Mexico and revealed that friendships and the social environment impact consumption decisions, often guiding students towards less healthy options due to social pressure and the perception of cost. This suggests that the perceived economic value is not only linked to the price of the products but also to social norms and expectations that can influence consumption intention [58]. This makes it clear that the pandemic modified consumption patterns and the perception of the economic value of food, which caused alterations in purchase intention [67]. Based on what was found, the following hypothesis is proposed:

**H8.** 
*Perceived financial value significantly influences the willingness to consume healthy food brands.*


Figure 1 shows a graphic representation of the theoretical model according to the hypotheses raised.

## 3. Materials and Methods

The study used a quantitative, non-experimental, cross-sectional design approach using a self-administered questionnaire [68]. An online survey was conducted through Google Forms, the link to which was published through the official social networks of a private university. This organization promotes an appropriate lifestyle through healthy eating and includes the Unión products factory. The survey was applied in Lima, Peru, between July and December 2023. In this way, the research focused on consumers who stated that they were consumers of the Unión brand, which is a factory of bakery products and other derived products whose value proposition is healthy eating.

A final sample of 515 participants was surveyed. They were selected based on the criteria of being over 18 years of age, residents of the study region, having at least completed secondary education, and agreeing to participate in the survey voluntarily. To participate in the survey, each consumer had to provide informed consent at the beginning of the online questionnaire. Thus, all participants knew that their participation was voluntary, that their data would be analyzed anonymously, and that they would be used exclusively for academic and research purposes.

The sample size was calculated using the inverse square root technique of [69] or PLS-SEM models [70]. A pilot test with 50 participants obtained a minimum path coefficient of 0.12. With a significance level of 5%, the initial calculation indicated that the minimum sample size should be 429 cases. However, it was decided that the sample was to be expanded to 515 participants to increase the robustness and precision of the results.

There were no missing data in the present study since the survey was administered digitally with a configuration that required all questions to be answered mandatorily before being sent. Furthermore, because all the scales used were Likert-type, no extreme values were identified that could be considered univariate or multivariate outliers, which is consistent with previous research highlighting the ability of Likert scales to minimize the presence of extreme values due to their ordinal structure and bounded ranges [70]. Therefore, it was not necessary to apply additional procedures for handling missing data or outliers.

Of the 515 participants, 65.05% were women, and 34.95% were men. Most were single (94.37%), with an undergraduate university education level (89.71%), and 51.65% received income of up to two minimum wages. Details of the study participants can be seen in Table 1.

### 3.1. Questionnaires for Measuring the Study Variables

To assess the Healthy Lifestyle (HLB) variable, this study applied the five-item scale developed by Küster-Boluda and Vidal-Capilla [71], which measures key aspects of healthy living, including dietary habits and physical activity patterns. To assess the perceived value of healthy branded foods, the questionnaire developed by Köse and Kırcova [44] was adapted to measure four distinct dimensions of perceived value: perceived quality value (PQV), which evaluates product attributes and performance; perceived social value (PSV), which assesses social impact and recognition; perceived emotional value (PEV), which measures feelings and affective states associated with the brands; and perceived financial value (PFV), which evaluates monetary aspects. Three items were used for each construct to ensure comprehensive measurement while maintaining questionnaire efficiency. Finally, seven items adapted from the questionnaire developed by Kumar et al. measured the willingness to consume healthy food brands (WCHFB). [72], which captures various aspects of consumption intention and behavioral predisposition towards healthy food brands. All measurement items were assessed using a 5-point Likert-type scale, where “1” means “Totally disagree” and “5” means “Totally agree”, ensuring consistency across all variables. 

Before the application of the questionnaire, a two-stage validation process was carried out. First, content validation was carried out by consulting three experts on responsible consumption, who reviewed the relevance and clarity of the items. This made it possible to confirm the suitability of the items in the Peruvian context. Subsequently, a pre-test was carried out with a pilot sample of 40 participants to evaluate the comprehension and appropriateness of the questions. The pre-test results confirmed the clarity and precision of all the items, since making changes to those initially proposed was unnecessary. In addition, Cronbach’s alpha coefficient was calculated, obtaining a value above 0.70, which indicated a high internal reliability of each study variable.

For optimal data collection, the digital questionnaire was divided into two sections: the first section presented the 24 items already mentioned in a structured sequence, and the second section consisted of questions related to sociodemographic data such as age, sex, marital status, educational level, and average monthly family income.

### 3.2. Statistical Analysis Process

The data were analyzed using two statistical software packages. First, IBM SPSS version 25 was used to examine the respondents’ demographic data. Additionally, it was used for preliminary data screening, including outlier detection and multivariate normality analysis [64]. Then, to evaluate the measurement model, discriminant validity, convergent validity, and reliability tests were conducted. These tests included the Fornell–Larcker criterion, cross-loadings assessment for discriminant validity, Average Variance Extracted (AVE) for convergent validity, and Cronbach’s alpha and composite reliability indices [73].

Subsequently, Smart-PLS version 4.0 software was used to test the conceptual model, applying a two-step approach that evaluated the measurement and structural models, as Hair et al. recommended [68]. To test the hypotheses, the partial least squares method (PLS-SEM), a comprehensive multivariate statistical analysis approach that simultaneously examines the relationships among variables in a conceptual model, was used [74]. This method, which involves three or more variables, facilitates multivariate analysis. Furthermore, PLS-SEM was selected in the present study because it supports memory-building [75]. The structural model analysis included assessing path coefficients for predictive relevance [68]. Following recommendations from previous consumer behavior studies, a bootstrapping procedure with 5000 subsamples was employed to evaluate the statistical significance of path coefficients and indirect effects [76,77].

## 4. Results

According to Table 2, the study constructs exceeded the acceptable threshold for reliability (α and CR > 0.70) and convergent validity (AVE > 0.50) [78]. Healthy lifestyle (HLB) showed an α of 0.809, CR of 0.823, and AVE of 0.566, with factor loadings between 0.661 and 0.819. Perceived emotional value reached an α of 0.947, CR of 0.948, and AVE of 0.904, with loadings between 0.942 and 0.958. Perceived financial value obtained an α of 0.928, CR of 0.929, and AVE of 0.875, with loadings from 0.932 to 0.938. The perceived quality value had an α of 0.955, CR of 0.955, and AVE of 0.917, with loadings between 0.949 and 0.964. The perceived social value showed an α of 0.952, CR of 0.953, and AVE of 0.912, with loadings from 0.951 to 0.960. The willingness to purchase had an α of 0.929, CR of 0.931, and AVE of 0.701, with loadings between 0.799 and 0.872.

Table 3 shows the evaluation of the discriminant validity of the study variables. In this case, the Fornell–Larcker criterion was used. According to this criterion, the square root of the AVE of each construct should be greater than the correlations between that construct and the others [79]. In the present study, the results confirm that the model’s constructs are differentiated. The square roots of each construct’s average variance extracted (AVE) are more significant than the correlations between that construct and the others, supporting their conceptual uniqueness. Healthy Lifestyle (HLB) has an AVE square root of 0.753. At the same time, the perceived values—emotional (PEV: 0.951), economic (PFV: 0.935), quality (PQV: 0.957), and social (PSV: 0.955)—also exceed this threshold, as does willingness to consume healthy food brands (WCHFB: 0.837). These results ensure that each construct measures unique aspects and does not overlap significantly with others, which reinforces the validity of the model and the reliability of subsequent analyses.

Table 4 and Figure 2 below show the contrasts of the hypotheses. The results of the analysis confirm that a healthy lifestyle (HLB) has a positive and significant influence on the perceived value of quality (PQV), social (PSV), economic (PEV), and functional (PFV), with coefficients ranging from 0.384 to 0.406, supporting hypotheses H1, H2, H3, and H4. which allows us to infer that consumers with healthy lifestyles perceive multiple benefits from consuming healthy foods. Likewise, hypotheses H5, H6, and H7 were confirmed, evidencing that PQV, PSV, and PEV positively affect the willingness to consume healthy food brands (WCHFB), with coefficients of 0.377, 0.124, and 0.270, respectively, and the perceived value of quality being the most determinant factor. However, the perceived functional value (PFV) did not present a significant relationship with the willingness to consume such foods (coefficient of 0.088; *p* > 0.05), which does not allow H8 to be accepted, indicating that consumers prioritize aspects of quality, social and economic benefits over functional ones when making decisions to consume healthy foods. These findings highlight the importance of designing strategies that highlight healthy brands’ quality, social impact, and economic value to encourage consumption.

## 5. Discussion

The purchase of products belonging to healthy food brands is increasing and gains momentum every time the individual learns or suffers the consequences of the imbalance in their lifestyle; from there, lifestyle acts as a key factor every time an individual intends to make a particular purchase, on top of the subjective assessment that he or she attributes to the products. Under this context, this study has analyzed the influence that a healthy lifestyle exerts on the factors of perceived value, which includes perceived quality, social value, emotional value, and economic value. According to the findings, in the study population, lifestyle influences perceived quality; to support this fact, previous studies have been identified that found that consumers with a healthy lifestyle present a high attraction towards the quality of the products they acquire, since this population has as a priority to obtain products with high standards in terms of product composition and health benefits [80,81]. In addition, another reference states that a healthy lifestyle maintains a particular link with quality as a symbol of security and confidence [82]. This means that a healthy lifestyle prioritizes health and well-being as long as the products maintain adequate quality standards that inspire safety and confidence in nutritional values.

Furthermore, the results of this study have shown that a healthy lifestyle influences social value. To support this statement, previous studies have been found that refer to consumers with a healthy lifestyle assuming greater responsibility for taking care of themselves, as well as their environment, and are even careful when choosing their food, because according to their worldview, they buy products aimed at contributing to sustainability and are aware every time they choose a brand [34,83], since it has been shown that this type of consumer cares a lot about acquiring products from brands that reflect a sense of social responsibility [84]. From there, its association with emotional value is delineated since individuals generate a special attachment and a strong feeling towards a specific brand since they are aware that the brand’s products not only allow them to maintain their health but also generate an experience of security and confidence that inspires tranquility [44,85].

Similarly, the results show that lifestyle influences economic value. This statement is consistent with previous research supporting the idea that price is a key factor in a healthy lifestyle. Consumer behavior also involves evaluating the benefits of a product and paying fairly for it [86,87]. One explanation for the different prices compared to similar products is that healthier foods require more outstanding care to avoid using pesticides or other mechanisms that do not support environmental protection, which is why their prices increase [88]. Thus, a reasonable price symbolizes a relationship between quality and price [89]. This means that the more expensive a product is, within its limits, the greater the benefits or attributes that contribute to a balanced lifestyle. Consequently, the population with a healthy lifestyle will have low price sensitivity since they are willing to invest in healthy foods.

Additionally, this research has evaluated the perceived value factors and their influence on the intention to consume healthy branded foods; specifically, this study has shown that perceived quality influences the intention to consume healthy branded foods. To support this result, it is essential to point out that one of the qualities of food safety is to guarantee the quality of the food; this means that the higher the quality standard, the greater the impulse of people to consume a particular product [90]; in this context, many companies have used quality as a strategic element to increase their sales since it is not surprising that the quality of the products encourages people to make a purchase [91,92]. Another study that supports the findings of this research states that, in the area of healthy lifestyle, healthy brands, nutritional value, and other elements associated with the preservation of health, consumer preferences are evolving according to new food realities, so their purchasing decision is consolidated according to the perception of product quality [93].

Likewise, the perceived social value significantly influences the intention to consume healthy branded foods. This finding is supported by the claim [94] that one of the purchasing behavior patterns is based on the social acceptance of a brand; this means that socially responsible attributes become an essential factor when making a purchasing decision [95]. Although meeting the expectations of all consumers is unlikely, the literature establishes that as long as a brand has a good reputation for its social impact actions, it will have greater acceptance in the market; therefore, the intention to purchase is intensified [96,97]. Similarly, this study has shown that emotional value influences the intention to consume healthy branded foods. This result is consistent with previous studies that establish that the experience with a brand can be so positive that the consumer feels satisfied, which leads to an increase in their intention to purchase due to the utilitarian determinants that the product presents. In this scenario, a special and lasting emotional bond arises between the consumer and the brand. The emotional value is consolidated when there are experience goods, which implies that the consumer, beyond evaluating the functional attributes of a product, also considers the sensation acquired each time he or she identifies a product that contributes to the feeling of well-being [98,99].

However, there is sufficient evidence to affirm that the economic value of food does not influence the intention to consume healthy food brands; this result can be explained by consumer behavior when he or she looks for a quality product that meets his or her expectations and prefers to purchase it, without the price being a determining factor for the purchase [100]. Although part of the literature establishes that price justice promotes the consumption of a certain product, adopting in the short term the intention to repurchase [101], there are contradictions to this statement that support the findings identified in this study; for example, the economic value moves to the background when consumers are more concerned about staying healthy and looking for products that favor their well-being. In this way, it is stated that consumers place more value on the attributes of a product in terms of quality and nutritional benefits [33,102].

Another study that agrees with the results of this study refers to the fact that consumers who are more willing to assume more expenses tend to belong to a higher socioeconomic level, such as business owners or retirees [103]; this statement represents a basis to explain the results obtained since, as detailed above, the study participants do not correspond to this group. This statement differs from recent literature stating that the current consumer’s behavior presents a special impulse due to the growing interest in healthy foods, since the awareness of these assumes an essential role in the demand for food options that contribute to health without price becoming a limitation. [104,105,106]. Given these statements, it is important to take into account that the results of this study and the contradictions identified could be a response to cultural differences and socioeconomic factors [107], even more so when considering that the cited studies that have evaluated the interaction of the study variables reflect diverse realities in different contexts, so the existence of discrepancies between them is not surprising.

### 5.1. Implications

The research results expand the literature on healthy food consumer behavior by revealing significant differences from previous studies. While prior research, such as that of Kim et al. [35] and Moser [36], suggested that economic value was a determining factor in the willingness to purchase healthy foods, our findings in the Peruvian market contradict this premise, showing that consumers prioritize quality, social, and emotional values over price. This has important theoretical implications for the perceived value model in emerging markets, suggesting that the hierarchy of values may vary significantly depending on the cultural and economic context. From a managerial perspective, these findings contrast with traditional marketing strategies that emphasize the financial accessibility of healthy products. For example, while Rincón-Barreto et al. [40] and Gantiva et al. [41] highlighted the importance of labeling focused on functional attributes and price, our results suggest that brands in the Peruvian market should reorient their strategies towards building emotional connections and social benefits, similar to what was observed in more mature markets as indicated by Bonilla et al. [48] and Kurnia et al. [7], but with a particular emphasis on local cultural values.

Since perceived quality significantly impacts purchase willingness, brands should focus on improving the perception of the quality of their healthy products. This could involve ensuring superior quality of ingredients, emphasizing nutritional value, and showing authenticity and transparency in the production process. The findings reveal that perceived emotional value and social value also have a considerable influence on purchase intention. Marketing campaigns should highlight the emotional benefits of healthy foods, such as personal well-being and emotional satisfaction, and the positive social connections these products can generate, such as environmental sustainability or supporting local economies.

Perceived economic value was not shown to significantly impact purchase willingness, suggesting that consumers interested in healthy foods prioritize quality social and emotional values over price. This implies that healthy food brands could focus less on cost-cutting strategies and more on promoting social value and quality.

Food companies can benefit from differentiating their products based on these non-economic values. By positioning their brands as socially and emotionally responsible, they can engage consumers who view healthy food consumption as an ethical and personal well-being choice. Public health initiatives should consider how a healthy lifestyle and perception of quality influence purchasing decisions. Educational programs could be designed to highlight these aspects and encourage consumers to value the quality and emotional benefits of healthy eating.

This study in the Peruvian market context can be replicated in other countries to check whether the same factors—quality, social value, and emotional value—have a similar effect in other cultures and markets. Since economic value did not influence the willingness to purchase in this study, it would be interesting to investigate whether different trends are observed in other consumer segments or other products within the healthy food sector concerning the impact of economic value.

### 5.2. Conclusions

The findings of this study demonstrate that a healthy lifestyle influences the intention to purchase healthy brands based on the perceived value of these products, such as the perception of the quality of health products, the perceived social value, the perceived emotional value, and the economic value. This research also demonstrates that perceived quality significantly influences the intention to consume healthy branded foods to the extent that they promote a healthy lifestyle.

The study demonstrates the importance of companies highlighting the attributes of healthy products to encourage purchases through a positive perception. This indicates that the emotional connection of healthy brands with customers who maintain or wish to have a healthy life must be encouraged. This implies that companies should not limit themselves to marketing healthy products at high prices. In the Peruvian context, a healthy lifestyle also positively influences purchasing healthy products, as Peruvian citizens tend to choose healthy brands when making regular purchases. Therefore, Peruvian companies must continue to market this type of product, considering the above factors that drive sales. In this sense, companies that market healthy food products should consider highlighting and promoting quality, social, emotional, and economic value. By living a healthy lifestyle, they will contribute to society’s progress.

These findings may have a greater geographic scope if the study was not only considered in a single country. In the same way, if the sample did not only have a scope of young university students who usually have weak purchasing power, the perception of purchase intention could also be more representative.

### 5.3. Limitations and Future Research

This study presents certain limitations that should be considered. First, the sample obtained through non-probabilistic convenience sampling shows a significant concentration of young participants up to 30 (93.79%) and women (65.05%). This concentration could bias the results since perceptions and behaviors towards healthy food consumption may differ by age and gender. To address this limitation, future research should expand the diversity of the sample, including older adults and a more significant proportion of men, to validate the generalizability of the findings. Likewise, comparative studies between age and gender groups would allow for identifying possible differences in the perception of the value and intention to consume healthy foods. A second limitation lies in the geographical scope of the study, which is confined to a single country. To address this restriction, it is crucial to conduct a cross-cultural analysis. This will help us understand how cultural differences affect the relationship between Healthy Lifestyle Behavior (HLB) and various types of perceived value. It underscores the importance of global perspectives in research and the potential impact on marketing strategies for healthy food brands in different markets.

For future research, the following lines of action are suggested. Conduct a longitudinal study to examine how value perceptions and purchase intentions evolve as consumers adopt healthier lifestyles. This could provide valuable insights into the formation and development of healthy consumption habits. Moderating variables such as age, gender, or income level should also be incorporated to analyze how these demographic factors influence the relationships proposed in the model. Motivations for consuming sustainable foods could vary significantly among different consumer segments. Finally, the sample should be expanded to include a greater diversity of age and socioeconomic groups, allowing for a more comprehensive and representative view of perceptions and purchasing behaviors concerning healthy food brands.

Future research could expand the current theoretical model in several promising directions. First, nested models that incorporate mediating variables between a healthy lifestyle and perceived values, such as health consciousness or nutritional knowledge, could be examined. Second, it would be valuable to explore the interrelationships between different types of perceived value, for example, how social value could moderate the relationship between quality value and consumption willingness. Third, the model could be enriched by including new constructs such as perceived brand authenticity, labeling transparency, or social media’s influence in forming perceived values. It would also be relevant to examine the moderating role of sociodemographic and psychographic variables, such as level of nutritional education or orientation towards sustainability. Furthermore, future studies could investigate how prior experience with healthy brands and brand loyalty interact with perceived values to influence consumption willingness and incorporate constructs related to resistance to changing eating habits and perceived barriers to adopting healthy eating. 

## Figures and Tables

**Figure 1 foods-14-00213-f001:**
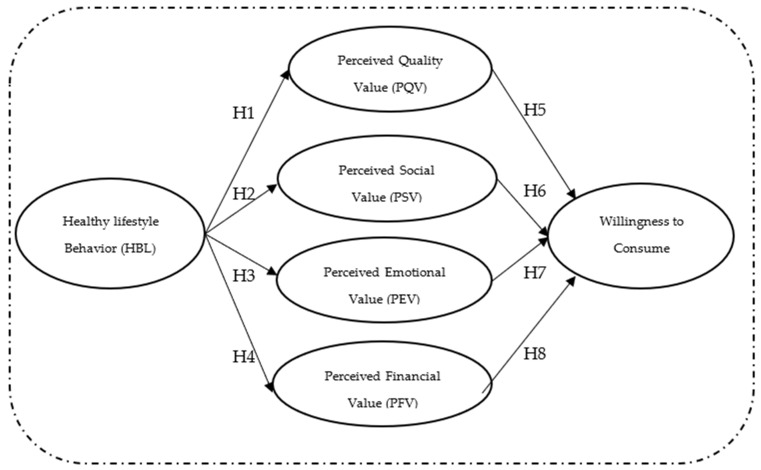
Conceptual model.

**Figure 2 foods-14-00213-f002:**
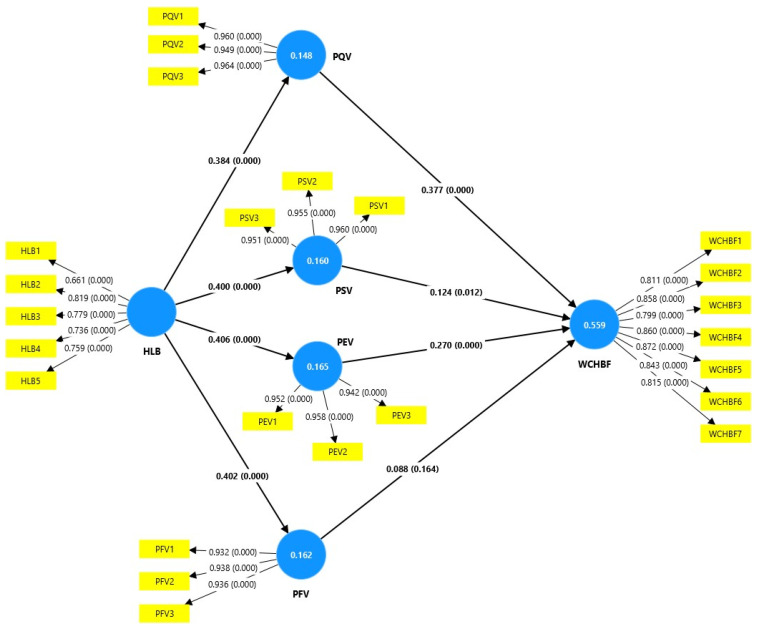
Structural model. HLB, healthy lifestyle; PQV, perceived quality value; PSV, perceived social value; PFV, perceived economic value; WCHFB, willingness to consume healthy branded foods.

**Table 1 foods-14-00213-t001:** Sociodemographic data (*n* = 515).

Demographic Item	Description	Frequency	Percentage
Age	Up to 20-years-old	256	49.71%
	From 21 to 30-years-old	227	44.08%
	From 31 to 40-years-old	20	3.88%
	41 years and older	12	2.33%
	Total	515	100.00%
Sex	Men	180	34.95%
	Women	335	65.05%
	Total	515	100.00%
Marital status	Single	486	94.37%
	Married	27	5.24%
	Divorced	2	0.39%
	Total	515	100.00%
Educational level	School	19	3.69%
	Technical	4	0.78%
	University (undergraduate)	462	89.71%
	University (postgraduate)	30	5.83%
	Total	515	100.00%
Family income	Up to 2 minimum wages	266	51.65%
	From 3 to 4 minimum wages	130	25.24%
	From 5 to 10 minimum wages	90	17.48%
	From 11 to 20 minimum wages	19	3.69%
	Greater than 20 minimum wages	10	1.94%
	Total	515	100.00%

**Table 2 foods-14-00213-t002:** Convergent validity results.

Variable	Code	Loading	Cronbach’s Alpha	CR	AVE
Healthy Lifestyle Behavior (HLB)	HLB1	0.661	0.809	0.823	0.566
	HLB2	0.819			
	HLB3	0.779			
	HLB4	0.736			
	HLB5	0.759			
Perceived Emotional Value (PEV)	PEV1	0.952	0.947	0.948	0.904
	PEV2	0.958			
	PEV3	0.942			
Perceived Financial Value (PFV)	PFV1	0.932	0.928	0.929	0.875
	PFV2	0.938			
	PFV3	0.936			
Perceived Quality Value (PQV)	PQV1	0.96	0.955	0.955	0.917
	PQV2	0.949			
	PQV3	0.964			
Perceived Social Value (PSV)	PSV1	0.96	0.952	0.953	0.912
	PSV2	0.955			
	PSV3	0.951			
Willingness to consume healthy food brands (WCHFB)	WCHFB1	0.811	0.929	0.931	0.701
	WCHFB2	0.858			
	WCHFB3	0.799			
	WCHFB4	0.86			
	WCHFB5	0.872			
	WCHFB6	0.843			
	WCHFB7	0.815			

**Table 3 foods-14-00213-t003:** Fornell–Larcker criterion for discriminant validity.

	HLB	PEV	PFV	PQV	PSV	WCHFB
Healthy lifestyle behavior (HLB)	0.753					
Perceived emotional value (PEV)	0.406	0.951				
Perceived financial value (PFV)	0.402	0.734	0.935			
Perceived quality value (PQV)	0.384	0.704	0.711	0.957		
Perceived social value (PSV)	0.400	0.620	0.582	0.438	0.955	
Willingness to consume healthy food brands (WCHFB)	0.385	0.677	0.626	0.684	0.508	0.837

**Table 4 foods-14-00213-t004:** Hypothesis testing.

H	Hypothesis	Original Sample (O)	Sample Mean (M)	T Statistics (|O/STDEV|)	*p* Value	Decision
H1	HLB -> PQV	0.384	0.386	8582	0.000	Accepted
H2	HLB -> PSV	0.400	0.402	9291	0.000	Accepted
H3	HLB -> PEV	0.406	0.408	9125	0.000	Accepted
H4	HLB -> PFV	0.402	0.404	9233	0.000	Accepted
H5	PQV -> WCHFB	0.377	0.377	6533	0.000	Accepted
H6	PSV -> WCHFB	0.124	0.123	2502	0.012	Accepted
H7	PEV -> WCHFB	0.270	0.269	3911	0.000	Accepted
H8	PFV -> WCHFB	0.088	0.090	1392	0.164	Rejected

## Data Availability

The original contributions presented in the study are included in the article, further inquiries can be directed to the corresponding author.

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
