# Peer review of "The Influence of Healthy Lifestyle on Willingness to Consume Healthy Food Brands: A Perceived Value Perspective"

_foods, 2025, doi:10.3390/foods14020213_

Round 1
Reviewer 1 Report
Comments and Suggestions for Authors
Dear Authors,
your paper is well structured, however I suggest to:
- report in Section 1 (Introduction) in a more extensive form your reserch's aims;
- in the Section 3 (Materials and methods) extend the description of the questionnaire design and the methods applied;
-in the Section 4 (Results) explain in a better way the findings of your analysis, especially the results obtained with the application of the Fornell-Larcker criterion (lines 286-291). The descrition of the hypothesis testing (Table 4) is too syntetic.
- report a new Section 6 (Conclusions) with the aim to syntetize the study's results and limitations.
Author Response
Dear Reviewer,
Thank you very much for your informed comments, which helped us so much in improving the manuscript. We appreciated the time you spent doing this and tried our best to address all your comments.
We hope that this revised version of the paper reaches the expected standard, worthy of publication in this journal.
A detailed list of answers to your comments and suggestions is reported below.
Many thanks for your time.
Best regards,

Reviewer 2 Report
Comments and Suggestions for Authors
In the Abstract, the authors could better detail the results and contributions of the study developed. In general, the Literature Review is well structured, although the theoretical support of the H1 hypotheses (not only in the relationship between the two constructs, but presenting a clearer definition of "perceived quality value", which is also present in H2 and H5. The constructs "perceived quality" or "perceived value" are generally addressed) and H7 can be embodied. In the "Materials and Methods" section, it would be important to better explain how the data were collected and whether there were univariate and multivariate missing and outliers (and, in this case, whether there was any type of treatment for these cases). Given that the final sample was predominantly composed of young people and women, could this have generated some bias in the research results? It would be important to comment on this. Before being applied, did the questionnaire undergo any validation and/or pre-testing procedure? The result of hypothesis H8 could be discussed by the authors in greater detail, making possible inferences. In "Implication", authors could compare the theoretical framework used in the research, or results from previous studies, to compare them to the research results, accepting both the theoretical implications and the managerial implications. The authors could deepen the suggestion for future studies, indicating models nested in the theoretical model tested, other relationships between the constructs evaluated and other constructs that could be tested in the future. The articles below may be able to help with the literature review, conclusions or suggestions for future studies:
Eberle, L., Milan, G.S., Graciola, A.P., Borchardt, M., & Pereira, G.M. Purchase intention of organic foods from the perspective of consumers. Management of Environmental Quality: An International Journal, 34(5), p. 1406-1423, 2023.
Respectfully, these are my comments and suggestions to the authors.
Author Response

(The authors gave the same response as above.)

Round 2
Reviewer 1 Report
Comments and Suggestions for Authors
Dear Authors,
now your manuscript is suitable for the publication
Reviewer 2 Report
Comments and Suggestions for Authors
Analyzing the adjustments or improvements made by the authors, I understand that the suggested recommendations were made satisfactorily.